# Influences of aging and mating history in males on paternity success in the red flour beetle *Tribolium castaneum*

Renya Kawakami[1], Kentarou Matsumura[2]*

1 Graduate School of Agriculture, Kagawa University, Kagawa, Japan, 2 Graduate School of Arts and Sciences, The University of Tokyo, Komaba, Tokyo, Japan

* matsumura@g.ecc.u-tokyo.ac.jp

**Data Availability Statement:** All relevant data are within the manuscript and its Supporting Information files.

## Abstract

The investment of males in reproductive traits is often associated with their age. For example, several empirical and theoretical studies have demonstrated that older males make greater investment in reproduction compared with younger males. However, with regards to post-copulatory sexual selection, male reproductive success might be influenced by decreasing sperm quality with male age and the interaction between aging and mating experience in males. However, only a few studies that investigated influences of male aging as well as male mating experience on their post-copulatory sexual selection. In this study, we investigated paternity success influenced by the post-copulatory sexual selection of males at different ages in the red flour beetle *Tribolium castaneum*. To investigate the effects of the mating experience, the paternity success of older males who had experienced multiple matings (mated male) was compared with older males who had not experienced mating (naive male). The results of this study revealed that paternity success was not affected by male aging. In fact, naive old males exhibited significantly higher paternity success compared with old males who had previously mated. These results suggest that an interaction between male aging and their mating experience affected their paternity success, but not male aging. Our study has demonstrated that male aging affects their reproductive success in a complex interaction of multiple factors in *T. castaneum*.

## Introduction

Sexual selection is one of the mechanisms that has a major influence on evolution [1, 2]. Sexual selection refers to any selection that can arise due to the differences in fitness, associated with nonrandom success in the competition for access to gametes for fertilization [3]. For example, to obtain mating opportunities with females, males often employ exaggerated weapon traits in male-male combats. Moreover, to persuade the female to accept mating, males often display gaudy ornamental traits and exhibit unique courtship behavior. These behaviors are displayed before copulation and are referred to as "pre-copulatory sexual selection" [4]. Conversely, the sperm of every male is not necessarily used during fertilization in polyandrous species,

**Funding:** This work was supported by JSPS KAKENHI Grant Numbers 22K20664 and 23K14264 to K.M.

**Competing interests:** The authors have declared that no competing interests exist.

resulting in male-male competition for egg fertilization inside the female, and this is referred to as "sperm competition" [5]. Since this occurs after mating, it is referred to as "post-copulatory sexual selection" [4]. Sperm competition leads to the evolution of traits that are beneficial in this competition. In post-copulatory sexual selection, "cryptic female choice" also plays an important role in addition to sperm competition as a male-male competition, and affects male paternity success [6–8]. Cryptic female choice refers to nonrandom paternity success bias by females during or after mating. However, it is very difficult to separate the effects of cryptic female choice from those of sperm competition in paternity success [8].

Males need to invest resources for reproduction during both pre- and post-copulatory sexual selection; however, the resource allocation is influenced by various factors and can vary. Therefore, understanding these factors is important for understanding sexual selection. Male aging is one factor that affects male investment in reproductive traits [9–11]. Males are often injured or die in male-male combat. Considering that early adult males are expected to obtain several reproductive opportunities in the future, they might prioritize their survival and decrease investment into costly traits for reproduction. On the contrary, older adult are expected to obtain relatively reduced reproductive opportunities in the future, and as a result might increase their investment in reproduction, even at the risk of threatening their own survival [9–11]. Thus, males need to make restrained investments in reproduction during early adulthood (young male) and increase this investment during late adulthood (old male), when their chances of reproduction dwindle [10]. A previously conducted theoretical study has reported that, in several cases, increasing investment in sexual traits with age is a stable evolutionarily strategy [12]. Older males in the broad-horned flour beetle species *Gnatocerus cornutus*, fight harder for females, in male-male combat, than younger males [13]. The old male butterfly *Hypolimnas bolina* is also known to fight longer than young males [11]. Moreover, the old male butterfly *Bicyclus anynana* is known to achieve more reproductive success than young males [10]. These results demonstrate that male investment in reproductive traits in pre-copulatory sexual selection increases with aging. Therefore, male investment in post-copulatory sexual selection as well as pre-copulatory sexual selection is predicted to increase with age.

However, male investment in pre-copulatory sexual selection could possibly differ from their investment in post-copulatory sexual selection. Male investment into post-copulatory sexual selection often negatively correlates with investment into pre-copulatory sexual selection in several animal species [14–17]. Therefore, if this resource allocation tradeoff is unaffected by male aging, then investment in post-copulatory sexual selection may decrease with male age as a consequence of male investment in pre-copulatory sexual selection increasing with their age.

Conversely, independent of male investment, it is possible that male fertilization success varies with male aging. For example, considering that the quality of gametes declines with age, indicating that it is important affected to evolutionary and ecological consequences on gamete performance and fitness [18]. In particular, age-associated accumulation of new mutations in the germline [19, 20] decreases the genetic quality of gametes [18, 21]. Therefore, regardless of the increase or decrease in investment by males, altered sperm quality caused by aging can reduce male fertilization success. Nevertheless, these previous studies [19, 20] focused on humans, and their applicability to other species is not known. A meta-analysis of aging with regards to ejaculatory traits [22] found no consistent trend across all organisms. In insects, the ejaculatory traits tend to improve with age in some species; however, some other species exhibit a decline in these traits, leading to varied results. These variations are thought to be influenced by the ecology and environment of the species. Furthermore, as mentioned above, male fertilization success is not only dependent on the changes in male factors (ejaculate

volume, sperm mobility, and sperm size), but also on cryptic female choice. Therefore, the actual fertilization success of males with aging must be investigated in various insect species.

Moreover, few empirical studies have reported that male fertilization success also depends on their mating experience. For example, among the older males belonging to the eastern mosquitofish (*Gambusia holbrooki*) species, males who experienced several copulations demonstrated increased paternity success (The proportion of offspring sired by a specific male among the offspring produced by a single female from multiple males), while males who had not experienced copulation showed reduced paternity success [23]. Thus, the interaction between aging and mating history of males can affect post-copulatory sexual selection. However, research focusing on the interaction between aging and mating history with regards to post-copulatory sexual selection in male insects, particularly with regards to sperm competition is lacking.

The objective of this study was to investigate the influence of male aging and mating history on male paternity success (The proportion of broods that a male parent can obtain through sperm competition in the red flour beetle *Tribolium castaneum)*. T. *castaneum* infests stored products [24]. Several aspects of its reproduction are well understood; thereby making it a major model organism in studies of post-copulatory sexual selection [25, 26]. *T. castaneum* inhabits high-density populations with both sexes indulging in frequent and promiscuous copulation [27–29]. During mating males transfer sperm into the female, and the females can store the sperms for several months within an elaborate spermatheca [30, 31]. Several previous studies have been conducted on post-copulatory sexual selection, such as sperm competition and cryptic female choice. Therefore, *T. castaneum* is an appropriate model for the investigation of post-copulatory sexual selection.

In this study, we investigated whether paternity success is affected by male aging. If investment in post-copulatory as well as pre-copulatory sexual selections increase with male aging, paternity success should increase with male aging. Conversely, if sperm quality decreases with male aging and if the male becomes disadvantaged during sperm competition and cryptic female choice, their paternity success should decrease with male aging. If a virgin male invests more in reproduction as they age, the paternity success of older males should be higher in virgin males. In this study, experiments were conducted to test these hypotheses.

## Materials and methods

### Insects

The beetle culture *of* the species *T. castaneum* used in this study was maintained in the laboratory, in a mixture of whole meal (Yoshikura Shokai, Tokyo) enriched with brewer's yeast (Asahi Beer, Tokyo). The organisms were maintained at 28°C with a 16 h photoperiod (lights on at 07:00, lights off at 23:00). A modified method described in a previous study was used for rearing the organisms [32]. The conditions mentioned above mimic the native environment of these beetles which are pests that infest stored grain.

Wild-type strain males and females which are brown in color, were used in this experiment along with mutant strain males which were black in color. The wild-type strain had been maintained at the Universities of Tsukuba and Okayama in Japan for approximately 35 years [32]. We focused on the sperm competition ability of males from this strain, which will hereafter be referred to as the "focal male." The mutant strain, which is a phenotype that is frequently used as a marker in sperm competition studies in *T. castaneum*, is homozygous for an autosomal, semidominant black body color allele [25]. The black mutant strain had been reared in conditions which were similar to those for the wild-type strain Okayama University for approximately 15 years. All the individuals used in the experiment were the youngest, at 10 days post-eclosion, considering the sexual maturation period of *T. castaneum* [32].

## Paternity success

For investigating the relationship between male aging and investment in post-copulatory sexual selection in *T. castaneum*, paternity success was measured in terms of the results of sperm competition and cryptic female choice. Though offspring can originate from multiple males, in majority of the cases, paternity success is usually acquired by the last male to mate [36]. Since the paternity success changes of the male that mated last could be easily tracked, this study focused on that male. In this experiment, a female belonging to the wild-type strain was sequentially mated with a mutant strain male and a focal male; following which the paternity success of the focal male that was the second to mate was investigated. The paternity success value of the second male has been referred to as P2, whereas the paternity success value of the first male is referred to as P1.

The focal males were divided into two types to distinguish between the effects of aging on paternity success from those of mating experience. Focal male 1 was subjected to experiments following 10 days of age after eclosion, after which they were allowed to mate after every 100 days. Focal male 2, was aged and not allowed to mate until it reached 300 days after eclosion (Fig 1).

## Age vs. paternity success

In the experiment using focal male 1, a virgin mutant black male (random age) and a virgin wild-type female (10-days-old and reared under monosexual condition until used for the experiment) were randomly chosen and placed into a circular plastic container (10 mm in diameter) along with food. The pair was allowed to mate freely for 24 h, following which, the black male was removed from the container, and one focal male 1 was placed into the container along with the female that had already mated with the black strain male. Similarly, the focal male 1and the same wild-type female was allowed to mate freely for 24 h. *T. castaneum* male beetles often fail to mate and therefore, the pair was placed together and allowed to copulate freely without controlling the number of copulations. Thus, two males were separately mated with one female. The exact number of copulations were not measured. Thus, the number of males that failed to mate at all times could be reduced by a considerable extent [33, 34]. After mating was complete, the female was isolated in a plastic container (45 mm in diameter, 10 mm in height) with sufficient food, and was allowed to lay eggs for ten days, for obtaining a sufficient number of offspring for the measurement of paternity success. The female lays approximately 10 eggs per day. The progeny (81 ± 2.2, mean number ± SE) were maintained at 28°C for 30 days. The adult body color of the progeny was scored to assign paternity success and generate P2 scores after they developed into adults. The focal male 1 that had measurable paternity success at 10 days of age was subjected to repeated paternity success measurements at 100, 200, and 300 days of age. The reproductive lifespan of *T. castaneum* is 0.5–1 year [35]; hence, the paternity success measurements were conducted up to 300 days of age. In cases where neither of the competing males sired offspring (i.e., female fertility was zero), the data were not considered for analysis as it could not be ascertained whether the copulations were successful and resulted in sperm transfer or storage, as females may also influence sperm retention. Around one fourth of the focal males 1 died before day 300, due to which, the number of focal males 1 who achieved paternity success at the age of 300 days was smaller compared with 10-day-old focal males 1. The sample sizes used for statistical analyses were 89 at 10 days, 81 at 100 days, 75 at 200 days, and 45 at 300 days, respectively. During the experiment period (from the eclosion of focal male 1 until its death), each focal male 1 was isolated in one well of a 48-well tissue culture plate (Cell Star, Greiner Bio-One, Kremsmünster, Austria) to prevent contact with other individuals, with a sufficient amount of feed for survival. The black males

**(A) Test for male aging**     **(B) Test for male mating experience**

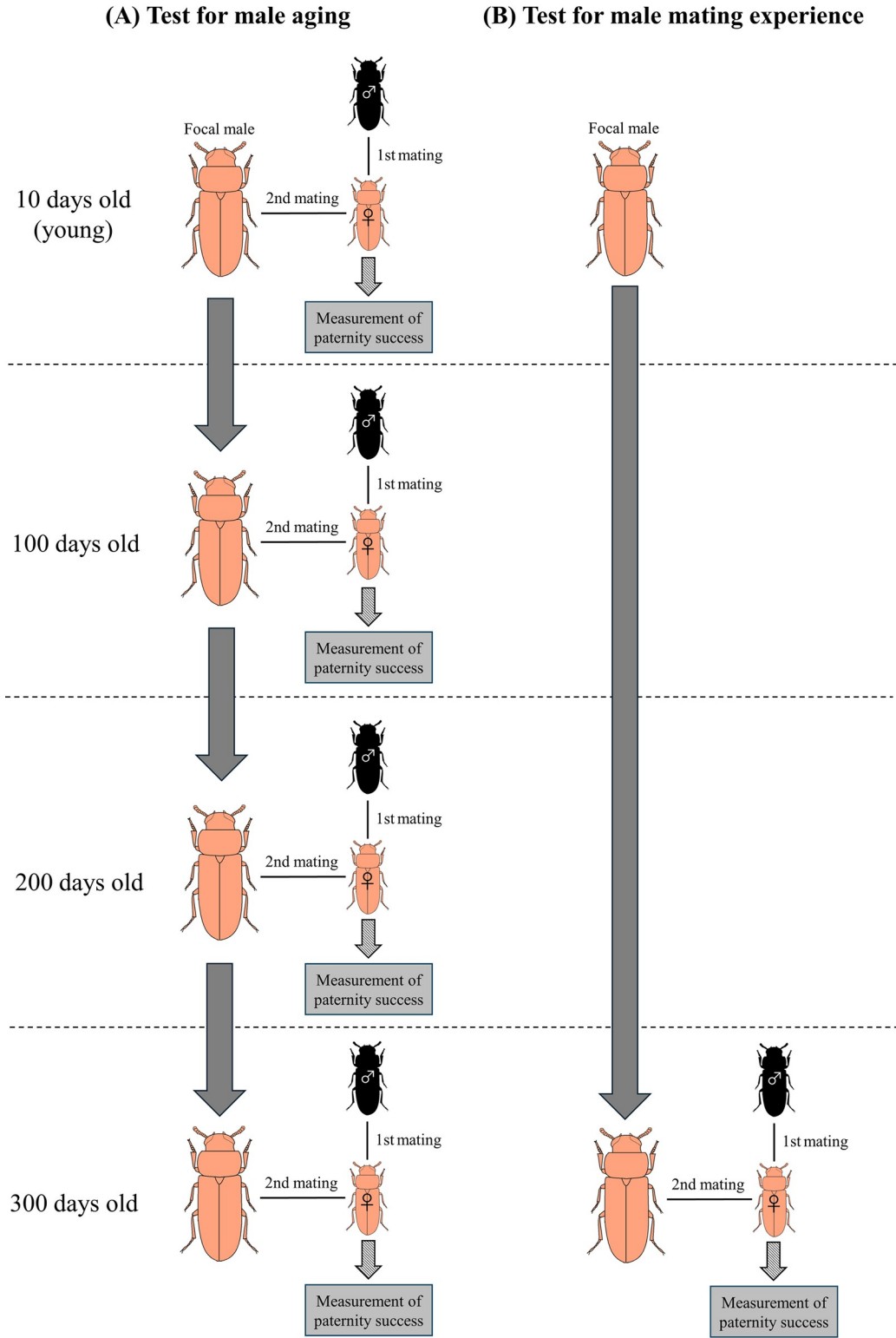

**Fig 1. Experimental design of this study.**

were reared under monosexual conditions with individuals of various ages. After the mating experiment, the black males used in the experiment were frozen and disposed. All experiments were performed between 12:00 hours and 18:00 hours in a room maintained at 28°C.

### Naïve male vs mated male

Paternity success of focal males 2, who had not experienced copulation until the age of 300 days was measured to investigate the effects of the interactions between aging and mating history on their paternity success. The focal male 2 was housed individually under the same conditions as focal male 1, during the experiment period (from the eclosion of focal male 2 until it reaches 300 days old). In this experiment, paternity success was measured using the same method described above. The sample size used for the statistical analysis was 102.

### Statistical analysis

Since our data were binary data with either wild-type or black beetles, a generalized linear mixed model with a binomial distribution was used to analyze paternity success. In this analysis, male age (10, 100, 200, and 300 days old) was an explanatory variable, and male ID was a random effect.

The comparison of paternity success at 300 days of age was conducted between males whose paternity success had been measured multiple times (mated male) and males whose paternity success had never been measured before (naive male); comparisons with the control males were conducted for both, 10-day-old and 300-day-old males, respectively. Approximately 46% of all males had died before completion of 300 days in this study. Thus, paternity success could be influenced by the differences in longevity among males. To test this effect, "paternity success at older age" was considered as the paternity success determined at the last measurement before the death of the focal male (i.e., if males die at the age of 150 days, the paternity success at 100 days old was used, and if males lived passed 300 days, the paternity success at 300 days was used). The paternity success at older age was compared with the paternity success at young age (10-days-old). A GLM with binomial distribution was used to analyze the results for the paternity success; however, it was over dispersed. Hence, the estimated dispersion parameter was used for correcting the variance [36, 37]. Bonferroni correction was used for conducting multiple comparisons, when a significant effect was detected in each test.

The software R ver.4.1.0 [38] was used to conduct all analyses, using the statistical packages *lme4* [39] and *car* [40].

## Results

### Age vs. paternity success

This experiment demonstrated paternity success data for males aged 10-days-old ($0.79 \pm 0.02$, mean ± SE), 100-days-old ($0.69 \pm 0.03$), 200-days-old ($0.74 \pm 0.04$), and 300-days-old ($0.65 \pm 0.05$). The results demonstrated a significant influence of age on paternity success ($F_{3,286} = 3.84$, $p = 0.0102$). Multiple comparisons with Bonferroni correction after this test revealed that there was no significant difference in paternity success at 10 days, 100 days, and 200 days, but at 300 days, the paternity success was significantly lower (Fig 2 and Table 1).

To exclude the effect of males that died prior to the completion of 300 days, the paternity success immediately before death was used, and a significant difference was observed when it was compared with the paternity success at 10 days ($F_{1,160} = 12.54$, $p = 0.0005$). Similarly, the paternity success of older males ($0.62 \pm 0.04$, mean ± SE) was significantly lower when compared with younger males (Fig 3 and Table 1).

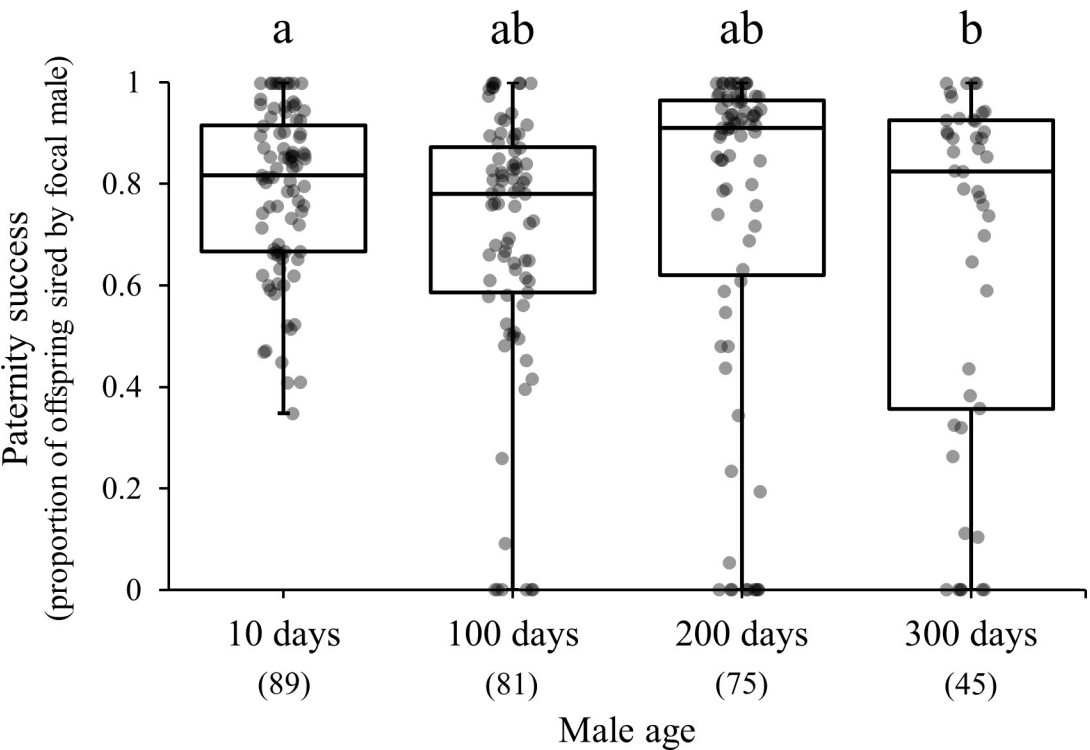

**Fig 2. Box plots for paternity success (The proportion of offspring originating from focal male 1 in the total offsprings produced by the female in each brood.) of *T. castaneum* males at the ages of 10, 100, 200, and 300 days, respectively.** Box plots depict the median and interquartile range of the data. Numbers below male age indicate sample size. There are significant differences among different characters (Bonferroni correction).

### Naïve male vs mated males

To exclude the influence of mating experience on paternity success, which was measured with male aging, "naive old males" that had never experienced mating until 300 days of age were included in the study and a comparison of the paternities of mated males and naïve old males revealed a significant difference ($F_{2,233} = 4.73$, $p = 0.0097$). The younger males demonstrated a significantly higher paternity success compared with the "naive older males" ($0.77 \pm 0.03$, mean $\pm$ SE) (Fig 4 and Table 1). However, compared to the older males that had experienced multiple mating until 300 days, naive older males exhibited a significantly higher paternity success (Fig 4 and Table 1).

Male longevity did not exert a significant effect on the lifetime paternity success of the male ($\chi^2_{1,79} = 0.20$, $p = 0.6521$; S1 Fig). However, male body size (precordial width) did exert a

**Table 1. Results of statistical analyses for paternity success.**

| Paternity success | Factor | df | F | P |
|---|---|---|---|---|
| 10~300 days | Age | 3 | 3.84 | 0.0102 |
| | Error | 286 | | |
| 10 days vs just before death | Age | 1 | 12.54 | 0.0005 |
| | Error | 160 | | |
| 10 days vs 300 days vs naïve male | Age | 2 | 4.73 | 0.0097 |
| | Error | 233 | | |

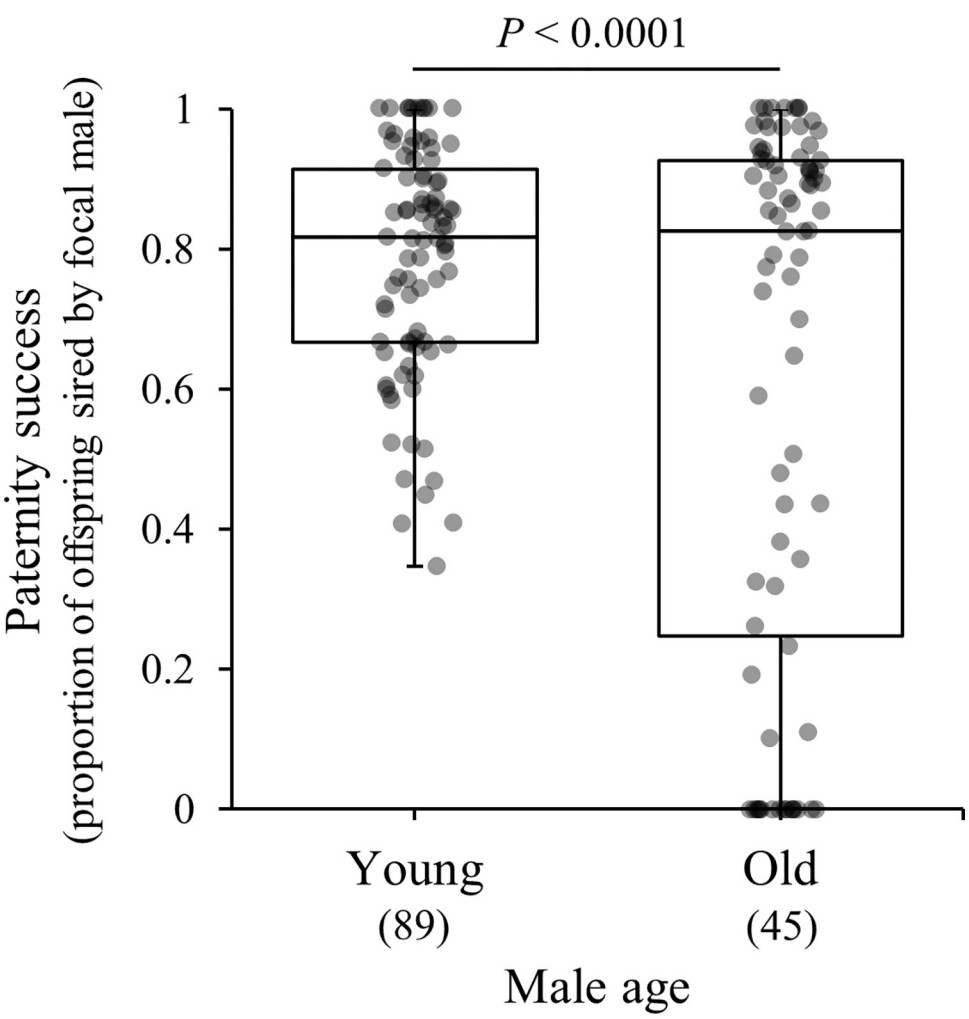

**Fig 3. Box plots of paternity success (The proportion of offspring originating from focal male 1 among the total offspring produced by the female in each brood.) of *T. castaneum* males young (10 days old) and old (prior to death at any age of days); for example, a focal male 1 that died at the age of 150 days had his paternity success measured at the age of 100 days, just before death.** Similarly, the paternity success of the focal male 1 who had died at the age of 250 days was considered as that measured at the age of 200 days, that is just before death. Box plots depict the median and interquartile range of the data. Numbers below male age indicate sample size.

significant positive effect on their paternity success ($\chi^2_{1,76}$ = 10.41, $p$ = 0.0013; S2 Fig). There was no significant correlation between body size and longevity in males ($t_{1,76}$ = −0.02, $p$ = 0.9856; S3 Fig).

## Discussion

This study investigated whether aging and mating experiences in *T. castaneum* males affected their post-copulatory sexual selection. Repeated measurements of paternity success at four different ages revealed that paternity success decreased with increasing ages of males and the number of measurements. However, a comparison of the paternity success of older males who had mated four times with that of older males who had not mated until the conduction of this experiment demonstrated that the paternity success of naive males was higher compared with mated males. Moreover, the paternity success of young virgin males and old naive males were

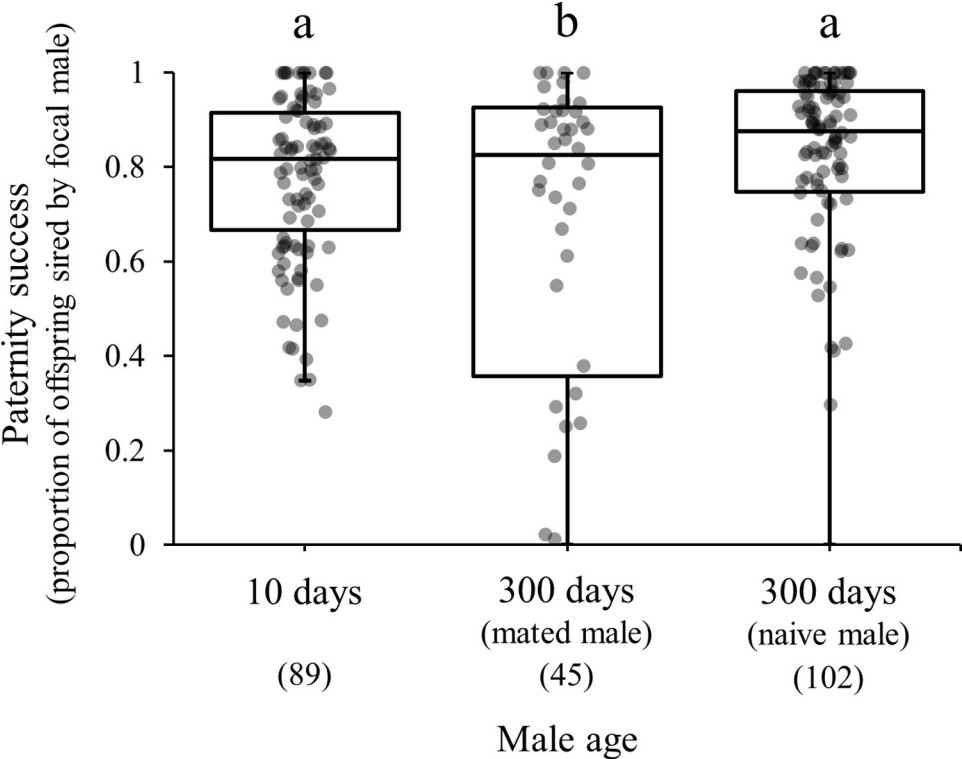

**Fig 4. Box plots depicting the paternity success of a young male (10-days- old), an old male with some copulation experience, and an old male without any copulation experience until measurement.** Box plots depict the median and interquartile range of the data. Numbers below male age indicate sample size. There are significant differences among different characters (Bonferroni correction).

similar. These results indicated that, while male aging did not influence post-copulatory sexual selection, older males who had previously mated sometimes showed a reduced investment in post-copulatory sexual selection. Thus, male paternity success is greatly influenced by the interaction between male aging and mating experience. The results of this study are particularly interesting as they differed from those observed in a previous study in the fish *Gambusia holbrooki* [23]. Although the fish species and insect species largely differ in terms of their *in vitro* fertilization *in vivo* fertilization respectively, this study is the first to detect increased paternity success in old males who did not experience copulation.

A previously conducted theoretical study had indicated that males can increase their fitness by increasing their investment in reproductive traits with age [11], and this has been demonstrated in pre-copulatory sexual selection in several species [10, 11, 13, 41]. However, our study did not observe any significant difference in paternity success between virgin young and virgin old males in *T. castaneum*, and as such cannot support the theoretical prediction. A meta-analysis conducted on post-copulatory sexual traits, specifically ejaculation traits of insect species, has indicated that ejaculation traits tend to improve with age [22]. However, several empirical studies have reported that in some animal species, males are disadvantaged with aging in post-copulatory sexual selection [42–46]. To clear the confusion regarding this issue, further empirical studies on several other species are warranted.

Several previous studies have indicated that decline in sperm quality with age leads to decreasing paternity success. Previous studies on humans have reported that the accumulation of harmful mutations in reproductive cells causes sperm quality to decline with age [18–20]. A

similar phenomenon might occur in insects as well. For example, it has been suggested that a decline in the seminal fluid quality due to aging leads to reduced paternity success in *Drosophila melanogaster* [47]. Furthermore, paternity success might also be influenced by the quantity of sperm transferred. Previous research has demonstrated that starved males of the *T. castaneum* species transfer fewer sperm per spermatophore compared to fed males, and have lower paternity success [48, 49]. This indicates the importance of sperm quantity for paternity success in *T. castaneum*. Therefore, the age associated decline in paternity success might be a result of the reduction in the quantity of sperm transfer. Although the details of the sperm quality and the volume of the ejaculate in association with male aging in *T. castaneum* could not be revealed in the current study, it is assumed that the effect of male aging on these ejaculate traits is minor, since the paternity success of naive old males was comparable with that of young males.

The results of this study, which demonstrated that paternity success did not differ between young and old males, was not in accordance with our hypothesis, according to which, investment of males in reproduction increases with their age. In several species, investment in pre-copulatory sexual traits has been reported to increase with male age [10, 11, 13]; Hence, it can be presumed that investment into pre-copulatory traits might increase with age compared with post-copulation sexual traits in *T. castaneum*. In a previously conducted study, *T. castaneum* males with high moving activity demonstrated increased mating success compared with males with low moving activity in [50]. It would be interesting to investigate the effect of male aging on the moving activity in *T. castaneum*. Moreover, if sperm quality decreased with the age of males, it was possible that cryptic female choice influenced the reduction in paternity success; however, our results did not support this possibility. It is possible that the quality of sperm does not deteriorate with age or that the effects of aging on male ejaculation are not recognized by females, but this has not been ascertained. Furthermore, although it was possible that the negative effects of male aging on mating behavior adversely affected paternity success, it has not been proven. Since mating behavior was not observed in this study, the details are not clear, and shall be examined in future studies.

Paternity success could be associated with male longevity. For example, males with shorter longevity might invest more in early reproduction, while those with longer longevity may invest lesser in early reproduction. However, no correlation between longevity and lifetime paternity success in *T. castaneum* males was observed in this study (S1 Fig). Thus, the paternity success observed in our study might not have been influenced by the differences in male longevity. However, male paternity success did depend on their body size (S2 Fig), indicating that larger males possess an advantage in achieving greater paternity success. This result indicates that either the larger males ejaculate more or that females prefer larger males; however, this cannot be ascertained from the results of this study. However, due to the lack of correlation between body size and longevity in males (S3 Fig), it can be assumed that male body size played a minor role on the results of this study.

A recent study reported that older males belonging to the *G. holbrooki* species, who had a greater experience of copulation exhibited higher paternity success compared with older males who were less experienced [23]. This suggests that male mating history affects their paternity success. However, the results of this study demonstrated a significantly higher paternity success of naïve males compared with males who had previously experienced multiple matings (Fig 4); thereby suggesting that paternity success is influenced by the interaction between their mating experience and male aging. Thus, our results contradicted the results obtained in the previous study [23]. Males belonging to the *G. holbrooki* species, exhibited fierce male-male combat for mating opportunities [51]. Thus, males who had extensive mating experience might have been preferred by females. In *G. cornutus*, the male possesses exaggerated weapon

traits, and it has been reported that females do not always prefer males with large weapons that dominate in male-male combats [52]. Conversely, males of the *T. castaneum* species do not engage in fierce male-male combat for females; instead, they actively engage in copulation [25]. This indicates that the effects of the interaction between mating experience and aging in males on their paternity success might differ across species. Male mating history might not be an indicator of male quality in *T. castaneum*. The results of this study have indicated that male mating experience influences their paternity success to a greater extent compared with male aging in *T. castaneum*. Males with copulation experience tend to increase their longevity as they age, by investing less in reproduction and more in survival. On the contrary, older males that have no copulation experience (i.e., their fitness is zero) are expected to increase their investment in reproduction, thereby achieving greater reproductive success. This might explain the results of this study, according to which, males who did not have the opportunity of mating until 300 days of age achieved higher paternity success compared with those who had mated before.

This study revealed the effects of male mating experience on paternity success in males of *T. castaneum*, a model organism for post-copulatory sexual selection. However, several factors could not be verified in this study. The exact cause of reduced paternity success of older males who had experienced multiple copulations remains unclear. For example, sperm quality and/or ejaculate volume may be compromised in older males who had experienced multiple copulations. It is advisable to conduct these investigations in the future to understand post-copulatory sexual selection.

## Conclusion

This study has demonstrated that, male paternity success (traits influenced by both sperm competition and cryptic female choice, either singly or in combination) in *T. castaneum* was not affected by aging. However, older males who had experienced multiple copulations experienced lesser paternity success, indicating that post-copulatory sexual selection is influenced by the interaction between aging and mating experience of males. Since paternity success differed in older males, depending on whether they had previously copulated or not, it can be assumed that reduced male investment in post-copulatory sexual selection, rather than aging, was the more influential factor in this result. Reduced volume of the ejaculate in older males who had experienced multiple matings, might have put them at a disadvantage in sperm competition with rival males along with cryptic female choice. The results do not provide detailed evidence for these possibilities, and future research is warranted. However, very few empirical studies have previously focused on the interaction between aging and mating experience in post-copulatory sexual selection, and hence our findings are crucial in understanding sexual selection.

## Supporting information

**S1 Fig. Relationship between longevity and their paternity success (mean).**
(DOCX)

**S2 Fig. Relationship between body size (prothorax width) and their paternity success (mean).**
(DOCX)

**S3 Fig. Relationship between body size (prothorax width) and their longevity.**
(DOCX)

**S1 File. All relevant data of this study.**
(XLSX)

## Acknowledgments

Dr. Yukio Yasui made valuable comments on this study.

## Author Contributions

**Conceptualization:** Kentarou Matsumura.

**Data curation:** Kentarou Matsumura.

**Formal analysis:** Kentarou Matsumura.

**Funding acquisition:** Kentarou Matsumura.

**Investigation:** Renya Kawakami, Kentarou Matsumura.

**Methodology:** Kentarou Matsumura.

**Project administration:** Kentarou Matsumura.

**Resources:** Kentarou Matsumura.

**Software:** Kentarou Matsumura.

**Supervision:** Kentarou Matsumura.

**Validation:** Kentarou Matsumura.

**Visualization:** Kentarou Matsumura.

**Writing – original draft:** Kentarou Matsumura.

**Writing – review & editing:** Renya Kawakami, Kentarou Matsumura.

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
