## [Decision Letter · Decision Letter 0]

4 Apr 2024

PONE-D-24-05330Influences of aging and mating history in males on their paternity in the red flour beetle Tribolium castaneumPLOS ONE

Dear Dr. Matsumura,

Thank you for submitting your manuscript to PLOS ONE. After careful consideration, we feel that it has merit but does not fully meet PLOS ONE’s publication criteria as it currently stands. Therefore, we invite you to submit a revised version of the manuscript that addresses the points raised during the review process.

Dear Authors,

Thank you for submitting your work to Plos one.

The manuscript has been reviewed by several peer reviewers, and based on their decision, I regret to inform you that your paper cannot be accepted for publication in the current form. Reviewers have raised critical concerns about data availability. Hence, data supporting results needs to provided and results of the study should be discussed in light of sexual selection and reproductive biology of insects and other animals. The entire manuscript also needs linguistic revisions for clarity. Figures need to be polished for clarity and statistical values. The reviewers' comments are listed below.

We look forward to receiving your revised manuscript.

Kind regards,

Basana Gowda G, PhD

Academic Editor

PLOS ONE

Journal Requirements:

https://onlinelibrary.wiley.com/doi/full/10.1111/j.1420-9101.2009.01889.x/full

https://academic.oup.com/biolinnean/article/112/1/67/2415841?login=false

In your revision ensure you cite all your sources (including your own works), and quote or rephrase any duplicated text outside the methods section. Further consideration is dependent on these concerns being addressed.

“This work was supported by JSPS KAKENHI Grant Numbers 22K20664 and 23K14264 to K.M.”

“Dr. Yukio Yasui made valuable comments on this study. This work was supported by JSPS KAKENHI Grant Numbers 22K20664 and 23K14264 to K.M.”

“This work was supported by JSPS KAKENHI Grant Numbers 22K20664 and 23K14264 to K.M.”

5. In the online submission form you indicate that your data is not available for proprietary reasons and have provided a contact point for accessing this data. Please note that your current contact point is a co-author on this manuscript. According to our Data Policy, the contact point must not be an author on the manuscript and must be an institutional contact, ideally not an individual. Please revise your data statement to a non-author institutional point of contact, such as a data access or ethics committee, and send this to us via return email. Please also include contact information for the third party organization, and please include the full citation of where the data can be found.

6. Please remove your figures from within your manuscript file, leaving only the individual TIFF/EPS image files, uploaded separately. These will be automatically included in the reviewers’ PDF.

Reviewers' comments:

Reviewer's Responses to Questions

**Comments to the Author**

1. Is the manuscript technically sound, and do the data support the conclusions?

Reviewer #1: Partly

Reviewer #2: Partly

Reviewer #3: Partly

Reviewer #4: Yes

2. Has the statistical analysis been performed appropriately and rigorously? 

Reviewer #1: I Don't Know

Reviewer #2: I Don't Know

Reviewer #3: Yes

Reviewer #4: Yes

3. Have the authors made all data underlying the findings in their manuscript fully available?

Reviewer #1: No

Reviewer #2: No

Reviewer #3: Yes

Reviewer #4: Yes

4. Is the manuscript presented in an intelligible fashion and written in standard English?

Reviewer #1: No

Reviewer #2: No

Reviewer #3: Yes

Reviewer #4: Yes

5. Review Comments to the Author

Reviewer #1: In their study titled ‘Influences of aging and mating history in males on their paternity in the red flour beetle Tribolium castaneum’ the authors test if there is an interaction between the effects of male aging and mating history on paternity success in the model insect species T. castaneum. They found that paternity success significantly decreased with age, but surprisingly old males that did not mate before had significantly higher paternity success compared to experienced old males.

Overall, I think this study uses a simple, but effective and elegant study design to address a very relevant question in sexual selection and reproductive biology. What is the effect of male ageing and sperm aging on paternity? Nevertheless, I believe that the manuscript could be substantially improved in clarity.

Unfortunately, the authors state: ‘data available on request from the author’. Stating this is against journals policy (for good reason, I think) and I believe that the manuscript should only be published if the authors have also published the underlying data (or given very strong reasons why this is not possible). ‘Data available on request from the author’ has been shown to be harmful praxis that often leads to inaccessible data (also see Tedersoo et al 2021).

Major comments

Abstract: I made some wording suggestions (see below) and I think that the results are currently hard to follow when only reading the abstract.

Introduction: Overall, the structure is good, but I think that the hypotheses could be more clearly defined and the expectations more thoroughly explained. I also believe that the prominent focus on male-male combat is a bit misplaced, as the data do not address this topic. I would argue, that instead it could be more clearly defined what effects sperm aging and aging in general might have on the different treatments.

Methods: The methods need some substantial clarifications for me. Please see my comments below. In particularly sample sizes are missing in the whole manuscript. Only giving dfs for tests is not sufficient in my opinion. Please give starting sample sizes and individual, final sample sizes for each treatment, preferably (also) added to the figures. I think it is also crucial to more clearly state how the competitors (i.e. black) individuals were aged and maintained. Finally, I have reservations concerning the egg laying period and particularly the censoring of unsuccessful focals in the first mating test (see detailed comments below).

Results: Though this is nicely concise, I think the results could be described in slightly more detail. While I like the presented figures it is hard to appreciate the data in the current presentation. I think giving means and/or percentages in reduction or increase in the results text would greatly help with that. In addition, a table with full statistics for the individual post-hoc comparisons presented in the figures, could be worthwhile (preferentially including effect sizes and/or means+error).

Discussion: The discussion nicely summarizes the main findings and is structured well. Nevertheless, I think that the hypotheses explaining the results are not all straightforward and fully discussed. Please see my detailed comments for thoughts and suggestions on alternative interpretations.

Minor comments

Short title: ‘Influences of aging and mating history on their paternity’ – not clear what ‘their’ refers to here. Actually, I think that the ‘their’ could be removed here and also in the main title.

L20: Change to ‘males’?

L24: Remove ‘to’?

L30ff: This seems contradictory to me: Did mating history affect paternity or not? If ‘naive old males showed significantly higher paternity success than mated old males’, then mating history did matter, right? Please clarify. I think one can only clearly understand what is meant here when reading the whole manuscript, but the abstract should be understandable without that.

L34: Change to ‘mating history affected paternity success’?

L40f: For me, natural selection is ultimately also about increasing reproductive success. Maybe have a look at Shuker & Kvarnemo (2021) for some suggestions how to define sexual selection (though I do not fully agree with their definition either …).

L46: Please add a citation for ‘premating sexual selection’.

L46: No only in polyandrous species not all sperm are used for fertilization. The vast majority of sperm do not fertilize any eggs. I think ‘not the sperm of every male is used …’ might be meant here.

L49: Long and confusing sentence. Please revise.

L53: The acronym ‘CFC’ is never explained. Also, it is only used twice in the manuscript. I would avoid the use of acronyms where possible.

L59: ‘One factor affecting male investment in reproductive traits is male aging. ‘ Please add a reference for this.

L60: Change to: ‘... die in male-male combat ...’?

L65ff: I think this might be true for potentially terminal investments, like in the example above, but certainly not for all male reproductive traits (like sperm production). Offspring produced in early life should be just as valuable as offspring produced in late life. I would even argue the opposite is often true: Aging and deaths will make reproduction late in life less likely. Hence, investing early in life has larger mean pay-off.

L88ff: I think the recent meta-analysis on sperm aging by Sanghvi et al. (2024) would be very relevant here. (I am not one of the authors.)

L111: Remove ‘to’?

L128ff: Long and confusing sentence. Please revise.

L129: Which ‘previous study’? Please cite.

L149: What kind of ‘wild-type strain’? What is its origin?

L153: Ages are in relation to eclosion date? So, 10 days means 10 days post eclosion? Please specify.

L155: ‘brown’ meaning wild-type, correct? Maybe call them ‘wild-type’ thought to clarify?

L158: Could you please specify this? Were all beetles from the same generation (hence age-synchronised at the beginning of the experiment)?

L168ff: Not sure why this is a reason not to control for the number of matings. Behavioural data would surely still be informative. Still, I think it is fair to say that there likely was at least one successful copulation in each pair.

L173f: Omit? Just because something has been done, it does not mean it is not problematic. I think is fine, but this sentence is not needed in my opinion.

L176: Why only 10 days? Do you think that letting them lay eggs for longer might change things? Is there any evidence that P2 changes over time?

L178f: I am not sure that excluding males that did not reproduce at day 10 from the dataset does not bias the data. At later points, males with zero offspring were still included, right? If success is compared over time this might bias data towards a higher mean offspring number at day 10 compared to later time points, I think.

L188: What about before the first trial? Were males separated early on? I ask because males have frequent same-sex copulations which might affect the results. For the same reason: Please specify how the competitors were kept. Where they also separated until trials? Also, where they always the same age or were as old as the focal? Was this the same competitor for each male or always an new one?

L188: Apologies, but I do not know what a ‘sell plate’ is. Could you please specify?

L189: Change to ‘… contact with …’.

L195: Related to comment above: Were these males kept individually until day 300?

L200: Were the model residuals normally distributed and not over-dispersed? Otherwise ‘quasi-binomial’ might give a more reliable fit.

L222: This sentence is more a legend then a results sentence. Maybe rather start with a summary of the most important result (like the next sentence)?

L222: Change to ‘… was a significant effect ….’. Please check whole section is written in past tense.

Line 224f/227f: Again, I think this sentence is not needed. Saying what the figure shows is the object of a figure legend.

L228: Please clarify what is meant by ‘treatment’ here.

L230f: Could you please illustrate this in more detail? How large was the difference between naïve old males and experienced old males? I would also be interested in the statistics on this, as this is very interesting and, to me, surprising.

Figures: I like the figures! But please add sample sizes here or elsewhere. Also, Bonferroni correction is mentioned here, but not in the Statistical methods, I believe.

L235: Omit: ‘As shown in Figure 1’? Personally, I would throughout avoid direct references to figures or tables in the discussion.

L244: Maybe mention that this study was done in a different species?

L249ff: As in the introduction, I find it surprising that the authors focus so much on this paper. It is a great paper, but in my reading deals foremost with advertisement-, not with sperm competition, traits. Hence, I do not think that the presented data are a good fit to test these models. Indeed, the data does more closely fit what I would predict under sperm aging effects (see e.g. Sanghvi et al. 2924).

L255ff: I am no sure about this analysis, I have to say. What matters in the end is each individual’s lifetime reproductive success. Maybe it would be more informative to look at the total reproductive success of all individuals and correlate that with individual survival (10, 100, 200 or 300 days)? This could show if lifetime reproductive success trades-off with lifespan.

L260: Again, see Sanghvi et al. 2024.

L268ff: And on the other hand …? Seems something is missing here.

L270f: ‘… suggesting that the effect on paternity was greater with number of copulations than with male aging.’ I personally do not think this is likely. 4 days of mating interactions over 300 days seems very low for a species like T. castaneum…

L273ff: This section is again on the general decline of P2 with age, right? Maybe clarify this, as I thought first this would be a continuation of the discussion on why naïve males did better then experienced males.

L291: I find these trade-off arguments not very convincing, given that pre-copulatory traits (particularly mobility) should also increase the number of - or chances for mating. This should also increase P2 and repeated mating should be beneficial in sperm competition as well as sperm quality or quantity per mating. I think the authors argument only strictly holds if the number of matings were equal for each trail, which was likely not the case. I think this could be discussed as a limitation.

L297f: Add references for cryptic female choice definition?

L301f: Here it would be crucial to know more about the competitor (i.e. black) male (see comments on methods). This only holds if competitors were all young males. Might one use different competitor ages in the future to test this hypothesis?

L310: Initially unclear what is meant by ‘performance’. Maybe substitute with ‘courtship behaviour’?

L335: Don’t think ‘thus’ is correct here.

L340ff: I am struggling to understand the mechanism for this. Are males that copulated before ‘exhausted’? Again, given the high mating rates in T. castaneum this seems unlikely to me. If this is really about differential investment, how are naïve males supposed to know when to invest more. In other words: How does a male know that he needs to invest most resources late in life compared to early? Is it not most likely for a male red flour beetle to mate hundreds or thousands of times before reaching the age of 300 days - so how did this plasticity in investment into post-copulatory traits evolve?

L344: Change to ‘… had higher paternity success …’?

L354: Change ‘sperm attack’ to ‘last-male sperm precedence’?

L356f: True, but maybe it is more interesting to finish with a sentence about what the present data can teach us? - They are interesting after all.

References

Sanghvi, K., Vega-Trejo, R., Nakagawa, S. et al. Meta-analysis shows no consistent evidence for senescence in ejaculate traits across animals. Nat Commun 15, 558 (2024). https://doi.org/10.1038/s41467-024-44768-4

Shuker DM, Kvarnemo C. The definition of sexual selection. Behav Ecol. 2021 Aug 7;32(5):781-794. doi: 10.1093/beheco/arab055.

Tedersoo, L., Küngas, R., Oras, E. et al. Data sharing practices and data availability upon request differ across scientific disciplines. Sci Data 8, 192 (2021). https://doi.org/10.1038/s41597-021-00981-0

Reviewer #2: The subject of the article is original and shall be published, however, there are some flaws and room to make it better and more easily understandable to the scientific readers.

There is a need to some English polishing in the article.

The abstract is clear and concise.

In the introduction there are some references needed that are missing, regarding some sentences.

When mentioning a species for the first time, the descriptors' name shall be added.

There is some concern on whether results from studies with different animals (for example, fishes, birds, humans) may apply to discuss the results obtained here. The suggestion would be to add the information on what animal the reference refers to, at least.

I suggest separating the objectives in two, and rewrite them in the end of the Introduction (1-influence of age; 2-influence of previous copulations, or as authors find better).

The methodology needs some rearrangement to be more clear, and delete repetitions. Please describe how you will measure paternity success as soon as you refer to this. It is not clear to the reader if the "proportion of offspring sired by the male" refers to number of broods or to number of insects in each brood, please clarify.

Lines 147 and 148 (with P2) are very confusing, please re-write this to be more clear. A scheme would be a nice addition, stating the beetles used (focal male1, male 2, females, and their age, and, non focal males).

The fact that these beetles are from laboratory rearing for 40 years may interfere with their behaviour? Would like that authors could discuss this a bit.

Also, could the mutant beetles have some type of limitation or advantage in terms of sexual behaviour compared to the focal ones? Was this previously tested by the authors or in the literature?

I suggest to arrange methodology and results within the 2 objectives set: 1) age vs success and 2) naive vs mated males (+ longevity/age).

A scheme or infographics would facilitate the understanding of the trials and its link to the objectives.

Regarding the statistical analysis, it misses the presentation of a more detailed numerical output of the model(s) done.

Data presentation with box and whisker plots is quite visual, but at least in the last one the differences are not clear, so I suggest putting a table instead, adding the statistical output similarly to the figure.

Could Figure 1 and 2 be combined? Just a suggestion.

The described part of sperm competitiveness does not seem to be in the results (between focal and non focal males), or it is not clearly explained.

In the end of the discussion there is a reference to P2 (already described) and to P1, but this would need a little more attention, for example, in the introduction or methods, as it is not clear here.

Some further suggestions are done in the attached document.

Regardless of all the observations I maintain that this is a very interesting subject that shall be published, however it needs major revisions.

Reviewer #3: The manuscript by Kawakami and Matsumura examines the effects of age on sperm competitive ability (offensive) of the red flour beetle Tribolium castaneum. The authors compared paternity success of young males (10 days-old) and old males (100, 200 and 300 days-old) by allowing the focal male to mate with a female for 24h after the female was first given 24h to mate with a competitor male. The authors also examined sperm competitive ability of naïve, old males (300 days-old).

The results suggest that old Tribolium castaneum males became less able to compete with young male sperm as they age. Several studies have demonstrated that old males are less able than their younger counterparts in several reproductive processes, including sperm competition, making the results reported here in agreement with other similar studies. However, I have some concerns about these results which I have outlined below.

1. We are given very little information about mating behaviors in this species, particularly in information that may help the reader better interpret the reported results. Is spermatophore quality affected by male age as ejaculate quality is in Drosophila (https://doi.org/10.1016/j.exger.2017.12.013)? Further, can females influence the process of sperm storage in this species? Each of these may affect sperm quantity that enter the spermatheca which can alter sperm competition outcomes.

2. How often do females remate? A quick glance at the literature suggests that females can mate multiple times within the first few hours after introduction of the male. Similarly, does male age affect their ability to court and inseminate females? Male mating frequency may decline with age. What is the mating frequency of young males compared to old males? If younger males’ mate at higher frequencies, then they may transfer larger sperm quantities, potentially making them better to sire more offspring. This could give the appearance of better sperm competitive ability, but this might be an indirect effect of the transfer of larger sperm quantities rather than a function of sperm quality.

Also, if males mate less frequently than young males (due to declines in copulation attempts/success with age), a larger proportion of males wold have P2 scores of 0 with increasing male age. The increasing P2 scores of 0 seen in figure 1 may be indicative of worse sperm performance, or increased failure to mate. In the hide beetle Dermestes maculatus, males are less likely to transfer sperm as they age (https://doi.org/10.1093/beheco/arl077). This needs to be addressed.

3. It is known if male age affects sperm quantity transferred? In the hide beetle Dermestes maculatus, old males that transfer sperm transfer similar quantities as young males (https://doi.org/10.1093/beheco/arl077). However, in other species such as Drosophila, males transfer less sperm with age (https://doi.org/10.1073/pnas.200905311). Interestingly, the age effect in Drosophila is only observed in males that had previous mating opportunities (similar to the experimental setup here), but is not seen in old, naïve males.

Thus, while it may be difficult to control for the number of copulations by a mating pair, it may be necessary potentially minimize indirect effects of male age that can complicate interpretation of reported results. Given that the focus is on sperm ability, aspects that can be influence paternity scores need to be controlled as much as possible.

4. It states in the ms that males that had paternity success at the initial mating were then mated every 100 days. Thus, it is possible to examine paternity success at each mating event? Do you see an overall decline with age? This may be indicative of a failure to mate with increased male age.

5. The authors need to clearly state their sample sizes (preferably in the figures). Also, was this performed with only a single cohort of males?

6. I do not understand Figure 2. The legend states that the “old” group comprises males “prior to death at any age of days”. Is this simply all the old males from Fig. 1 (that is, all males that lived to at least 100 days)? Or is it data from males that were mated at 100 or 200 days but did not live to the next mating age? This needs to be clarified.

Reviewer #4: The manuscript titled "Influences of aging and mating history in males on their paternity in the red flour beetle Tribolium castaneum" aimed to evaluate. paternity success of males at each age in the red flour beetle Tribolium castaneum. To investigate effects of mating history, they also compared the paternity success between older males who experienced some copulations (mated male) and older males who did not experience copulation (naive male). Because the authors affirmed that "the relationship between male aging and paternity remains unclear.

Studies related to the mating behavior of Tribolium castaneum have been extensively researched. Some articles in this area were found in my research and are not included in the discussion or Introduction.

I ask the authors what the added value of these studies is compared to previous ones? What significant contributions do they bring to the understanding of this species, for example, in population control, infestations, and mass rearing?

Before proceeding with a more detailed review, the authors should explain in the text what distinguishes this work from others.

A note: Tribolium castaneum is written in italics.

6. PLOS authors have the option to publish the peer review history of their article (what does this mean?). If published, this will include your full peer review and any attached files.

Reviewer #1: **Yes: **Lennart Winkler

Reviewer #2: No

Reviewer #3: No

Reviewer #4: No

---

## [Author Response · Author response to Decision Letter 0]

30 Aug 2024

Dear Dr. Basana Gowda G,

First of all, thank you very much for extending the deadline for submitting our revised manuscript. Thank you for giving us the opportunity to repost our manuscript (PONE-D-24-05330). We corrected the manuscript as much as possible to accommodate the helpful comments from the excellent reviewers. There is no doubt that our revised manuscript is better than before. We describe our response immediately after each reviewer's comment. Our corrections in the main text are shown in red. Please confirm.

5. Review Comments to the Author

Reviewer #1: In their study titled ‘Influences of aging and mating history in males on their paternity in the red flour beetle Tribolium castaneum’ the authors test if there is an interaction between the effects of male aging and mating history on paternity success in the model insect species T. castaneum. They found that paternity success significantly decreased with age, but surprisingly old males that did not mate before had significantly higher paternity success compared to experienced old males.

Response: Thank you for many useful comments to our manuscript. We corrected the manuscript as much as possible to accommodate the helpful comments from your comments. There is no doubt that our revised manuscript is better than before. We describe our response immediately after each your comment. Our corrections in the main text are shown in red. Please confirm.

Overall, I think this study uses a simple, but effective and elegant study design to address a very relevant question in sexual selection and reproductive biology. What is the effect of male ageing and sperm aging on paternity? Nevertheless, I believe that the manuscript could be substantially improved in clarity.

Response: Thank you for your appreciation of our research. The reasons for this finding are summarized in the end of Discussion (Lines: 421-435).

Unfortunately, the authors state: ‘data available on request from the author’. Stating this is against journals policy (for good reason, I think) and I believe that the manuscript should only be published if the authors have also published the underlying data (or given very strong reasons why this is not possible). ‘Data available on request from the author’ has been shown to be harmful praxis that often leads to inaccessible data (also see Tedersoo et al 2021).

Response: Thank you for your important comments. All data used in this paper will be made available to anyone as supplementary material.

Major comments

Abstract: I made some wording suggestions (see below) and I think that the results are currently hard to follow when only reading the abstract.

Response: Thank you for your useful comments to this Abstract. We corrected these sentences following your suggestions (Lines: 20-37).

Introduction: Overall, the structure is good, but I think that the hypotheses could be more clearly defined and the expectations more thoroughly explained. I also believe that the prominent focus on male-male combat is a bit misplaced, as the data do not address this topic. I would argue, that instead it could be more clearly defined what effects sperm aging and aging in general might have on the different treatments.

Response: We corrected some sentences related to male-male combat in Introduction following your suggestions (Lines: 42-146).

Methods: The methods need some substantial clarifications for me. Please see my comments below. In particularly sample sizes are missing in the whole manuscript. Only giving dfs for tests is not sufficient in my opinion. Please give starting sample sizes and individual, final sample sizes for each treatment, preferably (also) added to the figures. I think it is also crucial to more clearly state how the competitors (i.e. black) individuals were aged and maintained. Finally, I have reservations concerning the egg laying period and particularly the censoring of unsuccessful focals in the first mating test (see detailed comments below).

Response: We corrected some sentences in Materials and Methods (Lines: 148-262). We added sample size (Lines: 218-219, 236).

Results: Though this is nicely concise, I think the results could be described in slightly more detail. While I like the presented figures it is hard to appreciate the data in the current presentation. I think giving means and/or percentages in reduction or increase in the results text would greatly help with that. In addition, a table with full statistics for the individual post-hoc comparisons presented in the figures, could be worthwhile (preferentially including effect sizes and/or means+error).

Response: We corrected sentences in Results section, and we added mean and se of each paternity success (Lines: 264-294).

Discussion: The discussion nicely summarizes the main findings and is structured well. Nevertheless, I think that the hypotheses explaining the results are not all straightforward and fully discussed. Please see my detailed comments for thoughts and suggestions on alternative interpretations.

Response: We corrected some sentences in Discussion following your comments (Lines: 297-435).

Minor comments

Short title: ‘Influences of aging and mating history on their paternity’ – not clear what ‘their’ refers to here. Actually, I think that the ‘their’ could be removed here and also in the main title.

Response: Removed “their”.

L20: Change to ‘males’?

Response: Corrected.

L24: Remove ‘to’?

Response: Removed “to”. 

L30ff: This seems contradictory to me: Did mating history affect paternity or not? If ‘naive old males showed significantly higher paternity success than mated old males’, then mating history did matter, right? Please clarify. I think one can only clearly understand what is meant here when reading the whole manuscript, but the abstract should be understandable without that.

Response: Corrected.

L34: Change to ‘mating history affected paternity success’?

Response: Corrected.

L40f: For me, natural selection is ultimately also about increasing reproductive success. Maybe have a look at Shuker & Kvarnemo (2021) for some suggestions how to define sexual selection (though I do not fully agree with their definition either …).

Response: Corrected (Lines: 42-45).

L46: Please add a citation for ‘premating sexual selection’.

Response: Corrected (Line: 50).

L46: No only in polyandrous species not all sperm are used for fertilization. The vast majority of sperm do not fertilize any eggs. I think ‘not the sperm of every male is used …’ might be meant here.

Response: Corrected following your suggestions (Lines: 51).

L49: Long and confusing sentence. Please revise.

Response: Corrected (Lines: 55-60).

L53: The acronym ‘CFC’ is never explained. Also, it is only used twice in the manuscript. I would avoid the use of acronyms where possible.

Response: Corrected (Line: 57).

L59: ‘One factor affecting male investment in reproductive traits is male aging. ‘ Please add a reference for this.

Response: Added references (Line: 66).

L60: Change to: ‘... die in male-male combat ...’?

Response: Corrected following your suggestion (Line: 67).

L65ff: I think this might be true for potentially terminal investments, like in the example above, but certainly not for all male reproductive traits (like sperm production). Offspring produced in early life should be just as valuable as offspring produced in late life. I would even argue the opposite is often true: Aging and deaths will make reproduction late in life less likely. Hence, investing early in life has larger mean pay-off.

Response: As you say, the length of a male's life span may greatly affect paternity. So, we looked at the relationship between male longevity and paternity, the relationship between male body size and paternity, and the relationship between male body size and longevity. Added those results (Lines: 290-294) and discussion (Lines: 361-372).

L88ff: I think the recent meta-analysis on sperm aging by Sanghvi et al. (2024) would be very relevant here. (I am not one of the authors.)

Response: We added sentences (Lines: 104-105, 320-322).

L111: Remove ‘to’?

Response: Corrected.

L128ff: Long and confusing sentence. Please revise.

Response: Corrected.

L129: Which ‘previous study’? Please cite.

Response: Corrected these sentences (Lines: 139-146).

L149: What kind of ‘wild-type strain’? What is its origin?

Response: Corrected these sentences (Lines: 158-169).

L153: Ages are in relation to eclosion date? So, 10 days means 10 days post eclosion? Please specify.

Response: Corrected (Line: 185).

L155: ‘brown’ meaning wild-type, correct? Maybe call them ‘wild-type’ thought to clarify?

Response: We changed from brown to wild-type.

L158: Could you please specify this? Were all beetles from the same generation (hence age-synchronised at the beginning of the experiment)?

Response: Corrected (Lines: 158-169, L172-236).

L168ff: Not sure why this is a reason not to control for the number of matings. Behavioural data would surely still be informative. Still, I think it is fair to say that there likely was at least one successful copulation in each pair.

Response: We added sentence (Lines: 172-187).

L173f: Omit? Just because something has been done, it does not mean it is not problematic. I think is fine, but this sentence is not needed in my opinion.

Response: We removed.

L176: Why only 10 days? Do you think that letting them lay eggs for longer might change things? Is there any evidence that P2 changes over time?

Response: In this experiment, about 80 eggs were laid over 10 days. Therefore, it is considered that the number of eggs is not low. It would be interesting to know whether the P2 value changes over time, but in this experiment the spawning period was set to 10 days because of labor difficulties (Lines: 203-206).

L178f: I am not sure that excluding males that did not reproduce at day 10 from the dataset does not bias the data. At later points, males with zero offspring were still included, right? If success is compared over time this might bias data towards a higher mean offspring number at day 10 compared to later time points, I think.

Response: Males with zero offspring at the time of paternity measurement at 10 days of age are likely to be sterile. Conversely, males that do not have zero offspring when paternity is measured at 10 days of age are at least not sterile. Therefore, we did not use males with zero offspring at the time of paternity measurement at 10 days of age for subsequent experiments, but used males with non-zero offspring for subsequent experiments.

L188: What about before the first trial? Were males separated early on? I ask because males have frequent same-sex copulations which might affect the results. For the same reason: Please specify how the competitors were kept. Where they also separated until trials? Also, where they always the same age or were as old as the focal? Was this the same competitor for each male or always an new one?

Response: In the experiment, the males were isolated so as not to contact each other (Lines: 222-223).

L188: Apologies, but I do not know what a ‘sell plate’ is. Could you please specify?

Response: Corrected (Line: 221).

L189: Change to ‘… contact with …’.

Response: Corrected (Line: 223)

L195: Related to comment above: Were these males kept individually until day 300?

These males also were also kept individually (Lines: 230-236).

L200: Were the model residuals normally distributed and not over-dispersed? Otherwise ‘quasi-binomial’ might give a more reliable fit.

Response: As you say, the DP was calculated and it turned out to be overdispersion, since both models showed values of about 20. Therefore, we used the estimated dispersion parameter to correct the variance (Collett, 2002; Faraway, 2016) (Lines: 255-259, Fig. 1-3, Table 1).

L222: This sentence is more a legend then a results sentence. Maybe rather start with a summary of the most important result (like the next sentence)?

Response: We corrected sentences in Results section (Lines: 264-294)

L222: Change to ‘… was a significant effect ….’. Please check whole section is written in past tense.

Response: We corrected these sentences (Lines: 264-294).

Line 224f/227f: Again, I think this sentence is not needed. Saying what the figure shows is the object of a figure legend.

Response: We corrected these sentences (Lines: 264-294).

L228: Please clarify what is meant by ‘treatment’ here.

Response: We corrected these sentences (Lines: 264-294).

L230f: Could you please illustrate this in more detail? How large was the difference between naïve old males and experienced old males? I would also be interested in the statistics on this, as this is very interesting and, to me, surprising.

Response: We corrected these sentences (Lines: 264-294).

Figures: I like the figures! But please add sample sizes here or elsewhere. Also, Bonferroni correction is mentioned here, but not in the Statistical methods, I believe.

Response: We added sample sizes (Fig 2-4, 218-219), and sentences for multiple analysis (Lines: 257-259).

L235: Omit: ‘As shown in Figure 1’? Personally, I would throughout avoid direct references to figures or tables in the discussion.

Response: Corrected these sentences (Lines: 298-313).

L244: Maybe mention that this study was done in a different species?

Response: Corrected (Lines: 298-313).

L249ff: As in the introduction, I find it surprising that the authors focus so much on this paper. It is a great paper, but in my reading deals foremost with advertisement-, not with sperm competition, traits. Hence, I do not think that the presented data are a good fit to test these models. Indeed, the data does more closely fit what I would predict under sperm aging effects (see e.g. Sanghvi et al. 2924).

Response: We corrected Discussion (Lines: 298-435).

L255ff: I am no sure about this analysis, I have to say. What matters in the end is each individual’s lifetime reproductive success. Maybe it would be more informative to look at the total reproductive success of all individuals and correlate that with individual survival (10, 100, 200 or 300 days)? This could show if lifetime reproductive success trades-off with lifespan.

Response: We tested this additionally because, as you say, male longevity and paternity may be related. Added results (Lines: 290-294) and discussion (Lines: 361-372).

L260: Again, see Sanghvi et al. 2024.

Response: Added (Line: 322).

L268ff: And on the other hand …? Seems something is missing here.

Response: We corrected these sentences (Lines: 298-435).

L270f: ‘… suggesting that the effect on paternity was greater with number of copulations than with male aging.’ I personally do not think this is likely. 4 days of mating interactions over 300 days seems very low for a species like T. castaneum…

Response: Thank you for your important comments. It may be very important for a male to have even one successful mating. Indeed, there should be a large difference between males that have had only four mating opportunities but have already produced some offspring and males that have yet to produce any offspring.

L273ff: This section is again on the general decline of P2 with age, right? Maybe clarify this, as I thought first this would be a continuation of the discussion on why naïve males did better then experienced males.

Response: We corrected Discussion (Lines: 298-435).

L291: I find these trade-off arguments not very convincing, given that pre-copulatory traits (particularly mobility) should also increase the number of - or chances for mating. This should also increase P2 and repeated mating should be beneficial in sperm competition as well as sperm quality or quantity per mating. I think the authors argument only strictly holds if the number of matings were equal for each trail, which was likely not the case. I think this could be discussed as a limitation.

Response: We corrected Discussion (Lines: 298-435).

L297f: Add references for cryptic female choice

---

## [Decision Letter · Decision Letter 1]

4 Oct 2024

PONE-D-24-05330R1Influences of aging and mating history in males on paternity in the red flour beetle Tribolium castaneumPLOS ONE

Dear Dr. Matsumura,

Thank you for submitting your manuscript to PLOS ONE. After careful consideration, we feel that it has merit but does not fully meet PLOS ONE’s publication criteria as it currently stands. Therefore, we invite you to submit a revised version of the manuscript that addresses the points raised during the review process.

Though authors have attended to comments in the previous version. Howerver, still there concerns raised by reviewers. Authors need to carefully address them thoroughly.

We look forward to receiving your revised manuscript.

Kind regards,

Basana Gowda G, Ph.D.

Academic Editor

PLOS ONE

Journal Requirements:

Additional Editor Comments:

Authors have not carefully attended to the comments raised by reviewers. Address them thoroughly.

Reviewers' comments:

Reviewer's Responses to Questions

**Comments to the Author**

1. If the authors have adequately addressed your comments raised in a previous round of review and you feel that this manuscript is now acceptable for publication, you may indicate that here to bypass the “Comments to the Author” section, enter your conflict of interest statement in the “Confidential to Editor” section, and submit your "Accept" recommendation.

Reviewer #1: (No Response)

Reviewer #2: All comments have been addressed

2. Is the manuscript technically sound, and do the data support the conclusions?

Reviewer #1: Yes

Reviewer #2: Yes

3. Has the statistical analysis been performed appropriately and rigorously? 

Reviewer #1: Yes

Reviewer #2: Yes

4. Have the authors made all data underlying the findings in their manuscript fully available?

Reviewer #1: No

Reviewer #2: Yes

5. Is the manuscript presented in an intelligible fashion and written in standard English?

Reviewer #1: No

Reviewer #2: No

6. Review Comments to the Author

Reviewer #1: I would like to thank the Authors for their careful consideration of my comments and their replies.

I believe that the changes made improved the clarity of the manuscript, but I am afraid there are still instances, where there is work to be done.

I think that the reasoning in the Introduction and Discussion is not always convincing and that the theory for sperm aging and male investment into pre-copulatory traits, post-copulatory traits and immunity/longevity need to be discussed in more detail if mentioned so prominently. I am surprised that the Authors discuss male strategies for investing into reproduction and longevity without once mentioning the word ‘trade-off’. I feel a better theoretical basis is needed to discuss the present findings.

I particularly appreciate, that the Authors decided to publish their data. I think this is valuable and good practice. To make this even better, I think a legend explaining the different variables in detail would be most useful. Additionally, I think that the body size measurements (presented in the Supplement) are missing in the data set. Please add these as well.

Detailed comments

L26: Should be ‘… that investigated influences …’?

L34: Should be ‘… suggest that an interaction…’?

L42: Should be ‘Evolution is change in allele frequencies of a population over generations’? I don’t think ‘genetic frequency’ is the right term.

L53: Could be shortened to ‘This is called…’?

L67-69: Please revise. ‘…are expected to obtain …’ and ‘… decrease investment into costly traits for reproduction …’?

L139-146: This is helpful, I think! Please make clearer that these are hypotheses (e.g. ‘their paternity success should increase with’ or ‘… is predicted to increase with…’

L310: ’… used by fish …’ Should be ‘… in the fish …’?

L348: ‘pre-copulatory sexual selection may increase with age’ – Selection was not measured here and I think it is not meant. I think it should be ‘Investment into pre-copulatory traits’.

L350: ‘pre-copulatory sexual selection may increase with age’ - Maybe the Authors could give an example which traits they have in mind here?

L379: Should be ‘influenced by the interaction’?

L393-395: I am sorry, but I still don’t buy this argumentation. Survival and reproduction are interrelated and ultimately a combination of both makes up an individual’s fitness. Only survival without reproduction does not give any fitness returns. Likewise, only reproduction with little investment into survival is not a good strategy. A more nuanced discussion could help clarify this, I think.

The argumentation here boils down to a trade-off between reproductive traits and immunity. Still, the Authors fail to mention even the word ‘trade-off’ and do not seem to engage with the literature on this. E.g. What is the evidence that such a trade-off exists? For the present data it seems that there is no such trade-off (Fig S1). Did males in the treatment without matings until 300 days old survive better compared to the other treatment? I think that mating is just inherently costly to males in Tribolium (see Spatt 1980). One reason might be that they pass scarce resources (like water) to the female as a sort of nuptial gift with their ejaculate (see Moiron et al. 2022). Hence, the present results might not be a trade-off or a question of investment by the males, as the Authors seem to argue here, but a by-product of mating being costly.

Finally, I still think that the way the Authors argue, they make the assumption that males ‘know’ how many matings they can expect in the future and how long they might survive. But this information is not available to the individual, hence selection should favor an optimal strategy for a ‘mean Tribolium life-history’ and/or plastic adaptations depending on past experience.

L401-406: Frankly, I don’t think this is needed. I see no chance that resource availability might have influenced the results. Particularly as mating trials included food and were only 24h long, right?

L417-419: This statement seems very generic. Why is it a problem that they have been in the lab so long? What differences might one expect from a natural population?

References

Moiron, M., Winkler, L., Martin, O.Y. & Janicke, T. (2022). Sexual selection moderates heat stress response in males and females. Functional Ecology, 36, 3096–3106.

Spratt, E. C. (1980). Male homosexual behaviour and other factors influencing adult longevity in Tribolium castaneum (Herbst) and T. confusum Duval. Journal of Stored Products Research, 16(3–4), 109–114

Reviewer #2: I would like to thank the authors for the revision made, the manuscript is clearer now, and it is almost ready for publication.

There are some minor issues that shall be addressed. Indeed, it could be useful either to correct some grammar errors and correct some sentences. This shall be done either by another review to English or by having the manuscript reviewed by an English reviewer as suggested by the authors.

Some species descriptors are lacking (for ex. lines 77, 79, 80, 116, 331).

Sentences in lines 357-359 could be more elegantly written: "unclear and shall be examined in future studies".

Some doubts about lines 370-372 - firstly you refer to a "strong dependence", then to a "small effect", is it possible to make it clearer?

Sometimes you refer only to paternity, but I believe that "success" shall be added after that word (ex. line 377).

Congratulations to the authors!

7. PLOS authors have the option to publish the peer review history of their article (what does this mean?). If published, this will include your full peer review and any attached files.

Reviewer #1: **Yes: **Lennart Winkler

Reviewer #2: No

---

## [Author Response · Author response to Decision Letter 1]

6 Nov 2024

Dear Dr. Basana Gowda G,

Thank you for giving us the opportunity to repost our manuscript (PONE-D-24-05330R1). We corrected the manuscript as much as possible to accommodate the helpful comments from the excellent reviewers. There is no doubt that our revised manuscript is better than before. We describe our response immediately after each reviewer's comment. Reviewers pointed out that our English was poor, so we had an English proofreader correct the English in this manuscript. In this revised manuscript, our corrections are in red and corrections due to English proofreading are in blue. Please confirm.

Journal Requirements:

Response: We could not figure out which “retracted papers” you pointed out. Could you please tell us which one is the “retracted paper”?

Additional Editor Comments:

Authors have not carefully attended to the comments raised by reviewers. Address them thoroughly.

Response: We did everything possible to address the points raised by the reviewers. Please take a moment to review the revised manuscript.

Reviewers' comments:

Reviewer #1: I would like to thank the Authors for their careful consideration of my comments and their replies.

I believe that the changes made improved the clarity of the manuscript, but I am afraid there are still instances, where there is work to be done.

Response: Thank you for many useful comments to our manuscript. We corrected the manuscript as much as possible to accommodate the helpful comments from your comments. There is no doubt that our revised manuscript is better than before. We describe our response immediately after each your comment. We had an English proofreader correct the English in this manuscript. In this revised manuscript, our corrections are in red and corrections due to English proofreading are in blue. Please confirm.

I think that the reasoning in the Introduction and Discussion is not always convincing and that the theory for sperm aging and male investment into pre-copulatory traits, post-copulatory traits and immunity/longevity need to be discussed in more detail if mentioned so prominently. I am surprised that the Authors discuss male strategies for investing into reproduction and longevity without once mentioning the word ‘trade-off’. I feel a better theoretical basis is needed to discuss the present findings.

Response: Thank you for your important remarks. You are right, the amount of investment in reproduction and survival in males is often expected to be a trade-off. However, what we looked at in this study was whether males that had the misfortune of never being able to mate until old age would increase their investment in reproduction. Therefore, we did not use the word “trade-off” in our discussion because it was difficult to explain the trade-off in terms of a simple investment in reproduction versus investment in longevity (Lines: 390-417).

I particularly appreciate, that the Authors decided to publish their data. I think this is valuable and good practice. To make this even better, I think a legend explaining the different variables in detail would be most useful. Additionally, I think that the body size measurements (presented in the Supplement) are missing in the data set. Please add these as well.

Response: We are sorry that we forgot to provide the data of body size and longevity. We added these data to the supplemental information.

Detailed comments

L26: Should be ‘… that investigated influences …’?

Response: Corrected (Line: 26).

L34: Should be ‘… suggest that an interaction…’?

Response: Corrected (Line: 35).

L42: Should be ‘Evolution is change in allele frequencies of a population over generations’? I don’t think ‘genetic frequency’ is the right term.

Response: Removed this phrase.

L53: Could be shortened to ‘This is called…’?

Response: That sentence was corrected by English reviewer (Line: 54).

L67-69: Please revise. ‘…are expected to obtain …’ and ‘… decrease investment into costly traits for reproduction …’?

Response: Corrected (Lines: 68-70).

L139-146: This is helpful, I think! Please make clearer that these are hypotheses (e.g. ‘their paternity success should increase with’ or ‘… is predicted to increase with…’

Response: Corrected (Lines: 143, 145-146).

L310: ’… used by fish …’ Should be ‘… in the fish …’?

Response: Corrected (Line: 319).

L348: ‘pre-copulatory sexual selection may increase with age’ – Selection was not measured here and I think it is not meant. I think it should be ‘Investment into pre-copulatory traits’.

Response: Corrected (Lines: 360-361).

L350: ‘pre-copulatory sexual selection may increase with age’ - Maybe the Authors could give an example which traits they have in mind here?

Response: Corrected these sentences (Lines: 362-365).

L379: Should be ‘influenced by the interaction’?

Response: Corrected (Line: 395).

L393-395: I am sorry, but I still don’t buy this argumentation. Survival and reproduction are interrelated and ultimately a combination of both makes up an individual’s fitness. Only survival without reproduction does not give any fitness returns. Likewise, only reproduction with little investment into survival is not a good strategy. A more nuanced discussion could help clarify this, I think.

The argumentation here boils down to a trade-off between reproductive traits and immunity. Still, the Authors fail to mention even the word ‘trade-off’ and do not seem to engage with the literature on this. E.g. What is the evidence that such a trade-off exists? For the present data it seems that there is no such trade-off (Fig S1). Did males in the treatment without matings until 300 days old survive better compared to the other treatment? I think that mating is just inherently costly to males in Tribolium (see Spatt 1980). One reason might be that they pass scarce resources (like water) to the female as a sort of nuptial gift with their ejaculate (see Moiron et al. 2022). Hence, the present results might not be a trade-off or a question of investment by the males, as the Authors seem to argue here, but a by-product of mating being costly.

Finally, I still think that the way the Authors argue, they make the assumption that males ‘know’ how many matings they can expect in the future and how long they might survive. But this information is not available to the individual, hence selection should favor an optimal strategy for a ‘mean Tribolium life-history’ and/or plastic adaptations depending on past experience.

Response: As you say, using the same word "fitness" for males in two different situations was likely to cause unnecessary confusion. Males that have already copulated need not rush to reproduce, so investment in risky reproduction may decrease with age. On the other hand, older males that have never mated may invest even more in reproduction to avoid fitness zero. Thus, we have discussed the possibility that differences in mating experience significantly affect the relationship between aging and reproductive investment. We corrected these sentences (Lines: 389-416).

L401-406: Frankly, I don’t think this is needed. I see no chance that resource availability might have influenced the results. Particularly as mating trials included food and were only 24h long, right?

Response: Removed this paragraph.

L417-419: This statement seems very generic. Why is it a problem that they have been in the lab so long? What differences might one expect from a natural population?

Response: Removed this paragraph.

References

Moiron, M., Winkler, L., Martin, O.Y. & Janicke, T. (2022). Sexual selection moderates heat stress response in males and females. Functional Ecology, 36, 3096–3106.

Spratt, E. C. (1980). Male homosexual behaviour and other factors influencing adult longevity in Tribolium castaneum (Herbst) and T. confusum Duval. Journal of Stored Products Research, 16(3–4), 109–114

Reviewer #2: I would like to thank the authors for the revision made, the manuscript is clearer now, and it is almost ready for publication.

There are some minor issues that shall be addressed. Indeed, it could be useful either to correct some grammar errors and correct some sentences. This shall be done either by another review to English or by having the manuscript reviewed by an English reviewer as suggested by the authors.

Response: Thank you for many useful comments to our manuscript. We corrected the manuscript as much as possible to accommodate the helpful comments from your comments. There is no doubt that our revised manuscript is better than before. We describe our response immediately after each your comment. We had an English proofreader correct the English in this manuscript. In this revised manuscript, our corrections are in red and corrections due to English proofreading are in blue. Please confirm.

Some species descriptors are lacking (for ex. lines 77, 79, 80, 116, 331).

Response: Sorry, we cannot understand this description. Could you please provide more details?

Sentences in lines 357-359 could be more elegantly written: "unclear and shall be examined in future studies".

Response: We corrected (Line: 373).

Some doubts about lines 370-372 - firstly you refer to a "strong dependence", then to a "small effect", is it possible to make it clearer?

Response: "Strong dependence" was used because there was a significant positive correlation between body size and paternity, but it was not necessary to use "strong," so "strong" was deleted from the sentence (Line: 381). Because the length of longevity may significantly influence our results, body size could have influenced present study if it was correlated with longevity. However, since there was no correlation between longevity and body size, we considered that effect of body size on our results may small.

Sometimes you refer only to paternity, but I believe that "success" shall be added after that word (ex. line 377).

Response: We added “success” after paternity throughout the revised manuscript.

Congratulations to the authors!

Response: Thank you so much.

Kentarou Matsumura

---

## [Editor Report · Decision Letter 2]

4 Dec 2024

Influences of aging and mating history in males on paternity success in the red flour beetle Tribolium castaneum

PONE-D-24-05330R2

Dear Dr. Matsumura,

We’re pleased to inform you that your manuscript has been judged scientifically suitable for publication and will be formally accepted for publication once it meets all outstanding technical requirements.

Kind regards,

Basana Gowda G, Ph.D.

Academic Editor

PLOS ONE

---

## [Editor Report · Acceptance letter]

10 Dec 2024

PONE-D-24-05330R2 

PLOS ONE

Dear Dr. Matsumura, 

I'm pleased to inform you that your manuscript has been deemed suitable for publication in PLOS ONE. Congratulations! Your manuscript is now being handed over to our production team.

Kind regards, 

on behalf of

Dr. Basana Gowda G 

Academic Editor

PLOS ONE